# Getting Smarter about Smart Cities: Improving Data Security and Privacy through Compliance

**DOI:** 10.3390/s22239338

**Published:** 2022-11-30

**Authors:** Mudassar Aslam, Muhammad Abbas Khan Abbasi, Tauqeer Khalid, Rafi us Shan, Subhan Ullah, Tahir Ahmad, Saqib Saeed, Dina A. Alabbad, Rizwan Ahmad

**Affiliations:** 1FAST School of Computing, National University of Computer and Emerging Sciences, Islamabad 44000, Pakistan; 2Abbottabad Campus, COMSATS University Islamabad, Abbottabad 22060, Pakistan; 3Faculty of CIS, Higher College of Technology, Abu Dhabi 41012, United Arab Emirates; 4Center for Cybersecurity, Brunno Kessler Foundation, 38123 Trento, Italy; 5SAUDI ARAMCO Cybersecurity Chair, Department of Computer Information Systems, College of Computer Science and Information Technology, Imam Abdulrahman Bin Faisal University, P.O. Box 1982, Dammam 31441, Saudi Arabia; 6SAUDI ARAMCO Cybersecurity Chair, Department of Computer Engineering, College of Computer Science and Information Technology, Imam Abdulrahman Bin Faisal University, P.O. Box 1982, Dammam 31441, Saudi Arabia; 7School of Electrical Engineering and Computer Science, National University of Sciences and Technology (NUST), Islamabad 44000, Pakistan

**Keywords:** compliance, data breaches, information security policies and procedures, Personal Identity Information (PII), vulnerabilities

## Abstract

Smart cities assure the masses a higher quality of life through digital interconnectivity, leading to increased efficiency and accessibility in cities. In addition, a huge amount of data is being exchanged through smart devices, networks, cloud infrastructure, big data analysis and Internet of Things (IoT) applications in the various private and public sectors, such as critical infrastructures, financial sectors, healthcare, and Small and Medium Enterprises (SMEs). However, these sectors require maintaining certain security mechanisms to ensure the confidentiality and integrity of personal and critical information. However, unfortunately, organizations fail to maintain their security posture in terms of security mechanisms and controls, which leads to data breach incidents either intentionally or inadvertently due to the vulnerabilities in their information management systems that either malicious insiders or attackers exploit. In this paper, we highlight the importance of data breaches and issues related to information leakage incidents. In particular, the impact of data breaching incidents and the reasons contributing to such incidents affect the citizens’ well-being. In addition, this paper also discusses various preventive measures such as security mechanisms, laws, standards, procedures, and best practices, including follow-up mitigation strategies.

## 1. Introduction

In recent years, many cities have declared their intention to become smart cities, initiating smart city programmes and deploying smart city technologies. This led to a plethora of data-driven smart city solutions. The data exchange aims to provide better services and ensure a good quality of life for citizens. The use of smart applications raises several concerns regarding data security and privacy violations. In essence, involved sectors (critical infrastructures, financial sectors, healthcare, and Small and Medium Enterprises (SMEs)) are responsible for securely storing, managing, and processing huge amounts of data in their centralized and distributed repositories [1,2]. However, the stored data is prone to various security threats at rest and in motion [3,4]. This stored data comprises administrative, operational, and financial records, which include Personal Identity Information (PII). Industries are responsible for storing and sharing the PII of their employees and their customers. This PII raises various security concerns, such as trust, authentication, and data privacy, etc. Here, data privacy is the most important aspect, which is compromised by both inside and outside threats. These threats are solely responsible for modifying, fabricating, and even breaching data [5,6]. Adversaries use different tools, either available off-shelf or self-made, to attack data privacy by breaching into secure industrial facilities [7,8,9]. These industry breaches greatly impact various industrial assets, such as finance, trust, reputation, and people [10,11]. Figure 1 shows an inter-connected infrastructure of various sectors (SMEs, Finance, Critical Infrastructure, Public organizations, etc.), which expose an enormous amount of data to the attackers (insider and external). The attackers hence obtain a large attack surface, which eventually leads to an increasing number of data breach incidents.

A data breach is an incident in which data is compromised and disclosed to unauthorized or non-legitimate users [12,13]. This can be conducted deliberately or non-deliberately due to non-compliance with standards, regulations, policies, and best practices; moreover, malicious attempts, such as hacking and malware attacks, also result in serious data breaches [14,15,16,17,18,19,20]. Over the past few years, cyber-physical systems have faced data breaches, resulting in the loss of valuable information, reputation, trust, goodwill, well-being, and customer confidence [21]. These events result in certain physical, financial, and legal consequences [22,23].

Various studies indicate an alarming increase in data breach incidents and their impact in recent years. Studies demonstrate that the global average data breach cost was 6.4 percent in 2018 [24]. According to a recent survey of data breach incidents, the number of data breaches in the United States have been increased by 170 percent in the past decade, that is, from only 662 in 2010 to over 1800 in 2021 [25]. An average breach cost in US is USD 9.44 million, which is the highest in any country. Moreover, US tops the average breach cost for last 12 years; however, Brazil reports the highest percentage increase of 27.8% from USD 1.08 million to USD 1.38 million [26]. Similarly, a 2014 Filkins study reported that 47 percent of American citizens were affected by cyber crimes and 43 percent of companies reported information breaches. The escalation of these cyber crimes and information breaches are due to low budget allocation towards the IT sector, which is as low as six percent [27,28]. Moreover, in 2015, the Ponemon Institute found that the average cost per loss was around $154 on a global scale [29]. Furthermore, over 50 percent of small businesses were more likely to report physical breaches [30]. In addition, 70 percent of health sector data breaches, comprising electronic and digital information of the patients, were due to the healthcare providers’ negligence [31].

The importance of data breach incidents is evident by the following facts and events: ➊ About 700 data breach events were reported annually in United States, which is far less than the actual number [32]; ➋ Telstra (an Australian telecommunications and media industry) lost their 734,000 customers’ personal information following subsequent online publication [32]; ➌ In 2012, LinkedIn (a social network) reported about the publication of 6.4 million hash passwords on a web forum [33]; ➍ Organizations demonstrate that about 84 percent incidents were due to unauthorized access to data involving 97 percent confidentiality exposure [34]; ➎ A report by Privacy Rights Clearinghouse (an NGO in California to protect individuals privacy rights), more than 4400 data breaches exposed about 1 Billion records approximately [35]; ➏ In 2010, WikiLeaks showed 250,000 data leaks implications on many countries and firms, such as, critical information regarding American National Security, disclosing top classified military and diplomatic communications, conspiracies, trafficking, stolen properties, and espionage under Espionage Act 1917 [36]; ➐ In 2018, the largest social network, Facebook, deliberately shared 87 million users data with Cambridge Analytica for the American presidential campaign (https://www.reuters.com/article/us-facebook-privacy/facebook-says-data-leak-hits-87-million-users-widening-privacy-scandal-idUSKCN1HB2CM accessed on 20 November 2022); and ➑ in 2018, the leading taxi service Careem lost 14 million users’ data including name, email address, phone number, and trip data from the Middle East, North Africa, Pakistan, and Turkey (https://www.databreaches.net/dubais-careem-admits-to-data-breach-of-14-million-users/ accessed on 20 November 2022).

The impact of data breaches stated above necessitates focused strategies to overcome this problem, for which various laws, regulations, policies, procedures, and best practices are being enforced. These principles ensure information security by compliance to avoid data breaches that vary in impact among various industries and businesses [37]. The aforementioned principles are framed to provide a checklist for compliance, which includes: ➊ ISO 27001 standard, which is used to regulate information security and management systems for mitigating and minimizing data breaches by providing a mechanism for the identification, assessment, and controlling the risk of these incidents [38]; ➋ Payment Card Industry Data Security Standard (PCI DSS) is an industry-standard used by credit card service providers for protecting the cardholder data by practising the 12-prescribed requirements [39]; ➌ US Government has framed a regulatory framework called the Federal Information Security Management Act (FISMA), which is used to protect US Government Information Systems. Federal agencies are required to ensure continuous system monitoring and each agency receives annual grade on FISMA compliance [40]; ➍ An integration to FISMA, National Institute of Standards and Technology (NIST) provides the implementation guidance for Federal Information Processing Standard (FIPS 200) by addressing seventeen control areas for risk management [37,39,40]; ➎ Graham-Leach-Bliley Act 1999 is one of the major requirements for financial institutions known as the safeguard rule, which protects the customer’s information including confidentiality, security and integrity [41]; ➏ Financial institutions are bound to develop and apply an information security program under the Federal Trade Commission (FTC) [42,43]; and ➐ Health Insurance Portability and Accountability Act (HIPAA) 1996 [44], administered by the Health and Human Services (HHS) Department, provides two main rules to address security concerns, such as, privacy and security. The privacy rules address that only individual identity information can flow for research purposes, whereas the security rules address certain controls for electronically protected health information. Besides that, HHS guides entities on how to comply with security requirements [44,45]. The impact of compliance with these principles can be judged by the reports, such as the Romansky report, which concludes a 6.1 percent reduction in data breach incidents, and the Verizon 2010 report concludes that PCI compliant organizations are 50 percent less vulnerable to attacks [37,39,40,41,42,45].

Different studies and reports on data breaches have been presented, but no comprehensive and consolidated study on the impact of data breaches and prevention strategies is available. This paper aims to cover this shortcoming with the following contributions:Highlight various aspects of data breaches in various sectors of smart cities.Present a taxonomy to show the impact of data breaches in smart cities.Present multiple preventive measures and best practices to minimize data breach incidents; and also include the impact of non-compliance with these principles.Propose future directions in developing a continuous data breach risk assessment framework.

The remaining paper is organized as follows: Section 2 presents an overview of data breaches in various sectors, which is followed by a state-of-the-art review in Section 3, which classifies and presents various types of impact due to such breaches. Section 4 outlines various preventive measures to control data breach incidents. Section 5 presents the analysis summary and a discussion on future directions. Finally, Section 6 concludes the study.

## 2. Data Breaches in Smart Cities

The reliance of smart cities on computer-based systems, usually called Information and Communication Technologies (ICT), results in the mass generation of data, including sensitive and private data; therefore, data security becomes the main consideration of all these organizations. Organizations consider various threats to their data to keep it secure, including data breach analysis. Various sectors of Smart Cities [46] include critical infrastructures [47], healthcare [48], IT (including Management Information Systems) [49], education [50], and other non-technical firms, etc., which are facing data breach threats. The literature presented in the following sections provides a comprehensive impact of data breaches in cyber-physical systems.

In the context of **small and large business organizations** [18], the omnipresent threat of data breaches cannot be ignored. The threats to data are due to sophisticated malicious attacks, considered one of the biggest security problems in recent years. Cloud computing deployment in these organizations also brings new risks, including security breaches. However, to analyze these threats, Kamala D. Harris presented a report on California Data Breaches in 2012–2015 [30], in which the authors conclude that the main reasons for these malicious attacks are malware and hacking, which are rapidly growing. The retail sector is also struggling with these types of breaching incidents. Moreover, physical breaches, including the loss or theft of unencrypted data devices, also contribute to data breaching incidents. However, this attack has decreased by 10 percent from 2012–2015 [18,30].

In the **healthcare sector** [48], the major threat of data breach revolves around the personal and sensitive information of patients, which includes patient’s name, address, contact, medical history, etc. [31,51]. These attributes of patients invite criminals to use their sensitive information to perform criminal activities such as fraud, identity theft, marketing and selling in the black market, etc. [31,52]. Similarly, Stachel D. et al. in [53] also highlight data breach as one of the major concerns in the healthcare industry in 2015. To address the prevailing issue, the authors applied an Actor-Network Theory (ANT) approach to describe the Protected Health Information (PHI) complexities and vulnerabilities. They breached data, which is the source of major breaches in the healthcare industry. ANT clearly describes the relationship and role of actors, i.e., individuals, systems, and groups involved in healthcare data sharing and the complexities involved in the healthcare environment. The study determines different associations between data breaches, data storage locations, business acquaintances, covered entities and individuals being affected [53].

In the **gaming industry** [54], Choi et al. [32] and O. Hinz et al. [55] provided studies of various data breach incidents in small and big gaming firms, such as Sony PlayStation (P. Station. https://www.playstation.com/en-us/ accessed on 20 November 2022), Electronic Arts (EA) (E. Arts. https://www.ea.com/ accessed on 20 November 2022), Sega (SEGA. http://www.sega.com/ accessed on 20 November 2022), Nintendo (Nintendo. https://www.nintendo.com/ accessed on 20 November 2022), Ubisoft (Ubisoft. https://www.ubisoft.com/en-gb/ accessed on 20 November 2022) etc. The aforesaid firms were attacked to extract their users’ information, which these firms manage on their website for the online selling of games and other benefits. The studies contribute to the market for understanding various breaching incidents and their attack vector.

Similarly, the studies also provide information about a breach incident on the Target Corporation (a US-based Retailing company) that was used by millions of users for online buying of games and other necessary things. The company’s repository was hacked, which ultimately compromised millions of customers’ data and sensitive information. This incident ultimately resulted in crimes such as identity theft, online fraud by phishing, spamming, Denial of Service (DOS), password cracking, and backdoor Trojans. The company loss may exceed millions of dollars due to no provision and implementation of an Information Security (IS) model and controls capable of recovering from such incidents [32,55].

The most serious incidents of data breaches are considered in the critical and cyber-physical systems [46,47], such as government sectors, banks, and public sector departments [56]. The recent event of a data breach in one of the leading banks, i.e., JP Morgan bank, which even has implemented high-security measures, and standard compliance procedures, but still faced data breaching incidents, which is one example of data breaches even in the critical but secure sector [35]. The new technological advancements and offer customer services in the banking sector, such as the provisioning of electronic banking or e-banking facility, also face new security challenges and risks; according to Naema et al. [57], while e-banking facilitates many e-commerce activities, it ultimately leads to analyzing, collecting, and storing customers’ sensitive information. Apart from many opportunities, this new approach is also liable to privacy, security, and trust issues. As a result, data breaches in the banking sector negatively affect customers’ satisfaction with e-banking due to weak security controls, which keeps the customers in a constant threat of monetary loss.

Similarly, Julian Assange’s WikiLeaks disclosure provides a huge amount of intelligence about critical sectors and cyber-physical systems around the globe. These leaks also provide critical Information regarding American National Security, which can be exploited by the attackers. Moreover, the targeted leaks containing sensitive information were subsequently published on a public website, which made a huge physical and financial impact on the critical sectors around the globe [36].

## 3. Impact on Assets

The preceding section briefly highlights the gravity of data breaches in various sectors. Every incidence of a data breach has some impact, which ranges from minimal to very grave. Understanding these various impacts from their type and consequence in perspective is important.

This section presents a classification of data breaches, their impacts, and preventive strategies. According to Robert Layton and Paul A. Watters [33], an organization comprises financial assets and people. Here, financial assets are further divided according to the literature on four parameters, namely: (a) cost, (b) revenue, (c) market value, and (d) stock price. Next, the asset people are further classified as (a) customers and (b) employees, which impact the organization in terms of loss of trust, reputation, turnover, and confidence. A state-of-the-art taxonomy is presented in Figure 2, which is further enriched in the following sections (also see Tables 4–7).

### 3.1. Financial

The financial impact includes all costs incurred due to a data beaching incident, which comprises costs in investigating and identifying policy, cost of hiring staff, cost of restoring deleted data, communication costs, audit and compliance costs, and revenue costs, including unavailability, insurance, and use of stolen information [33].

The existing body of research finds many studies to estimate data breaches’ economic and financial impact. The financial and economic impact of security breaches can be estimated in terms of cost and revenue [19,26,58,59,60]; Hirschprung et al. estimated the cost and value of maintaining personal information in information systems [60]. Similarly, Martin et al. presented the idea of quantifying the data breach impact on revenue due to the non-availability of resources and rendering the employees idle; consequently, the security attack results in the seizure of various business processes, which ultimately decreases the revenue generation [58]. Ponemon Institute, together with IBM, have been conducting various data breach studies to estimate the average breach costs to last many years. The average cost considers many cost factors such as legal, regulatory, technical activities, loss of brand equity, customer turnover and drain on employee productivity. The results of these studies demonstrate how the potential cost of information breach incidents is increasing annually, which ultimately causes a great loss to various businesses and society. In the latest report published in 2022, the average cost of a data breach in US has reached a record high of $4.35 million, which represents a 2.6% increase from the last year ($4.24 million) and a 12.7% increase from 2020 when it was $3.86 million. Furthermore, various breaches were reported publicly, but unfortunately, the cost associated with data breach incidents did not reduce. The main reasons for these breaches were increased hacking, malware and privilege misuse by activist groups and insiders [19,26,58,59,60].

The impact on financial assets due to a data breach also includes the organization’s market value and stock price [34,61,62]. This impact is difficult to measure and estimate its true value. However, a hypothetical approach is adopted to estimate the breach’s impact on market value and stock price. The research demonstrated weak statistical results and failed to conclude that publicly reported breaches either had no or little impact on subsequent targets in terms of their stock price and market value. The result also weakly concluded that both market and organizations react similarly to subsequent data breach incidents [61].

Moreover, Sotirios Pirounias et al. also provided a mechanism to investigate the impact of information breaches on a firm’s market value. The study demonstrated the impact of cyber attacks for which firms spent over 1 trillion dollars on the IT sector. The authors presented this impact for all publicized breaches. The statistical results demonstrated that about 0.33 percent of the firm’s market value was lost daily, with an average loss of 168 million dollars per breach event. To deal with the aforesaid problem, the authors suggested that the firms must allocate more budget to the IT sector [34].

### 3.2. People

The stakeholders of an organization or its people, either its valued customers or employees, are also important assets affected by data breach incidents. The major factors which impact these assets are not tangible but cannot be ignored. For example, Schatz et al. in [61] studied the impact of repeated breaches on a single organization, affecting reputation and customer trust (non-tangible assets). The subsequent breaches eliminated the trust factor of a customer in the organization, as they believed their information and sensitive data was not safe. As a result, organizations lose a customer’s trust, turnover, and acknowledgement [61]. Similarly, Choongy et al. also studied the negative impact of data breaches, which badly affected the market’s view, resulting in a loss of customers, trust, and reputation [28].

While the impact of a data breach is on the people, it is also important to study the role of people in data breach incidents, which is investigated by various researchers [30,31,32,52,63]. According to a study [52], people are one of the weakest entities in the cybersecurity chain because they intentionally or unintentionally become a source of data breach incidents. Similarly, researchers in [30,31] found that people are prone to errors, unintended exposure to the public internet, and negligence. These reasons have an overall impact of 17 percent, among which half of the impact is on the government sector. Moreover, 70 percent of healthcare organizations compromise electronic or digital patient information. This loss of information by the people, according to [63], allows criminals to obtain Personal Identity Information (PII) such as name, address, social security number or credit card number. The PII is so critical that its loss results in identity crime and other serious security threats for vulnerable people and/or organizations. Floyd et al. analyzed such breaches in their study [52] and found that the breached PII of the customers was sold for a very low cost (approximately $1.25) which was then used for malicious activities, such as marketing in the black market. The impact on people due to such breaches and public exposure of PII was studied by Choi et al. [32], which explained how people were affected by data breach incidents by crimes such as identity theft and online fraud. These crimes were performed by phishing, spamming, DOS, password cracking, and backdoor Trojans, which ultimately lost millions of dollars [32].

## 4. Preventive Measures for Data Breach Incidents

To overcome and mitigate data breaches and their impact in various sectors, various preventive measures are proposed, which can be categorized as three main approaches described in the next sections; these include (a) security mechanisms, (b) standards, policies, procedures, regulations and best practices, and (c) legal obligations.

### 4.1. Security Models and Techniques

The first preventive measure to mitigate data breach incidents is adopting certain models and techniques, generally known as security mechanisms. While security mechanisms is a broader term used to refer to various security techniques (e.g., encryption, digital signature, etc.) to achieve security services (e.g., confidentiality, integrity, availability, etc.), this section specifically refers to models and techniques used for the prevention of data breaches. The literature provides various models and techniques, such as Data Leakage Prevention (DLP) [64,65,66,67], Information Security (IS) model based on the theory of justice and psychological contract framework [32], economic and financial models [33,58,59], Actor-Network Theory (ANT) [53], and privacy and access control technique [68].

The use of DLP systems [64,65,66,67] can protect and prevent the breach or leakage of data at rest, in use and in motion. According to to [65], to detect and prevent data breach incidents, various approaches are adopted in DLP systems, such as content analysis, virtualization and isolation, policy and access rights, cryptographic techniques, social and behaviour analysis, quantifying and limiting, data identification, data mining and text clustering, advance criteria for characterization, and statistical analysis. Different approaches offer different potentials and challenges; for example, the content analysis approach uses data fingerprinting and regular expression, which has the limitation of not correctly perceiving the changing semantics of confidential data. This can be addressed by using statistical analysis, which can effectively manage any data that has variations; therefore, a data leak prevention system with statistical analysis capability can effectively approximate the existence of data semantics [65]. The authors in [67] also proposed a statistical data leakage prevention (DLP) model that classifies the data based on semantics. The results demonstrate that the proposed statistical DLP system could correctly classify documents even in cases of extreme modification. It also had a high level of precision and recall scores [67].

Similarly, the authors in [64] proposed a theoretical advance criteria approach for characterizing breach incidents and their relevance to attack. The proposed characterizing criteria help analyse breach incidents to identify various loopholes, which can be overcome by significantly improving information security measures. The proposed criteria for identifying breaches consist of elements such as actors, intention, access points, access mode, identity, access rights, data classification, data state, information value, communication channel, communication medium, communication mode, incident detection, and recurrence risk [64]. Similarly, Choi et al. [32] presented an Information Security (IS) model to prevent data breach incidents in gaming firms. The model is based on the theory of justice and the psychological contract framework. The proposed framework utilizes three main types of justice, such as, distributive, procedural, and interactional. Based on these justices, the framework can represent customer behaviour, justice perceptions, and psychological responses. The framework generates the aforesaid responses from the time of breach incident to the firm’s recovery from such breaching incidents. Therefore, the model provides researchers with an effective conceptual framework to reduce and mitigate the effect of breach incidents on customer relationships.

Researchers have also proposed models to reduce economic and financial impacts due to data breach [33,58,59]. The authors in [58] present an idea to quantify the financial impact of non-productivity and employees’ idle time resulting from the non-availability of IT services after a cyber-attack. The authors propose a theoretical model that provisions alternative work tasks independent of the availability of IT resources, which can eventually decrease employees’ idle time. Algarni and Malaiya also propose an economic model [59] with strong prediction capability for calculating the cost associated with data breaching incidents. The model is also capable of analyzing possible reasons for breaching incidents. The proposed models for cost assessment and identifying reasons for breaches eventually help organizations analyse overall breach costs and effects to overcome subsequent incidents. The authors in [33] presented a model taken from the Office of the Australian Information Commissioner (OAIC) to estimate the tangible cost of a data breach event. The presented model consists of four steps, namely: (a) containing the breach, (b) evaluating the risk, (c) notifying the breach, and (d) mitigating the impact of a breach. The authors also applied their proposed model to two case studies, the Telstra breach in 2011 and the LinkedIn breach, to calculate the cost associated with these breaches. The results demonstrate that Telstra and LinkedIn stock prices increased due to the notification of breaches and improved security services to avoid future attacks such as phishing, scamming, malware, and insiders.

Stachel et al. address the increasing concerns of a data breach in the healthcare industry and apply the Actor-Network Theory (ANT) approach to describe the complexities of Protected Health Information (PHI), which is sensitive and vulnerable information resulting in healthcare industry’s data breach incidents [53]. The authors present the following list of recommendations for the avoidance of data breach incidents: (1) each entity or actor involved in a healthcare environment must be trained, accountable, and auditable; (2) the human actor engineered breaches are sorted, so that the actor can be held accountable for illegal and access violation activities; (3) vendor management should be improved by enforcing vendors to implement better data security policies; (4) notification of breach to other actors should be incorporated in order to help other actors preparing for subsequent breaches to limit the damage; (5) compliance with the existing healthcare industry policies and standards, developed under HIPAA or Centers for Medicare and Medicaid Services [69], must be ensured including continuous audit; (6) ensure privacy with encryption of data while transmitting through any source, storing, and retrieving from databases; and finally (7) the external factors are not always the key for data breaches; rather, the internal factors are also critical. The authors, therefore, conclude that in addition to ensuring the security of devices, the security of data itself is also very important. Hence, the organisation’s Chief Information Security Officer (CISO) must be empowered to improve the technology, processes, and physical environment [53].

In addition, the authors in [70] proposed a model based on certain factors such as level of exposure, security and other organizational factors. The model was developed to check whether data breaches in the context of healthcare industries occurred or not. The dataset used in the study was taken from the United States Department of Health and Human Services (DHHS) and the Healthcare Information & Management Systems Society (HIMSS), comprising 1804 breaching incidents. The model used on the aforementioned dataset was tested on binary logistic regression to examine the data breaches’ impact. The result demonstrates that the used factors were associated with data breaching incidents in the healthcare industry but with a high error rate. Moreover, the model presented was in its initial state of data breach modelling that could be improved by taking more factors into account [70]. Finally, the authors in [68] proposed a privacy and access control scheme to protect patient information in the healthcare information system. The paper proposes an e-TRON architecture, which allows patients to delegate access rights to the medical staff by assigning them delegation tokens. In this way, patient information access can be better controlled by the patients themselves [68].

### 4.2. Standards, Regulations, Policies, Procedures, and Best Practices

Information breach prevention is not always based on effective risk management practices but can also be mitigated by complying with industries’ or governments’ framed regulations and standards. The authors in [37] examine the effectiveness of regulations applied in different industries to analyze whether the information breach impact is reduced. Information security regulations and standards to avoid data breaches vary among industries and businesses. These framed standards and regulations provide a checklist for compliance. To understand how standards and regulations can provide prevention, the literature provides an overview of different standards, such as ISO 270012013: [38], PCI DSS [39], FISMA with FIPS 200 [40], and HIPAA administrated by HHS [37,44].

The International Organization for Standardization (ISO) (https://www.iso.org/ accessed on 20 November 2022) publishes various standards, including a family of information security management system standards in ISO 27K series. The main standards ISO 27001:2022 and 27002:2022 provide best practices and controls for the *establishing*, *implementing*, *maintaining* and *continually improving* Information Security and Management System [71,72]. Additionally, the standard also specifies requirements for risk management and a holistic list of 93 security controls divided into four chapters/domains, shown in Table 1. The latest ISO 27001:2022 version introduces 11 new controls (compared to the previous 27001:2013 version) including threat intelligence, data masking, data leakage prevention and information deletion, which specifically focus on data breach prevention.

For banking and other financial institutions [56], compliance with Payment Card Industry Data Security Standard (PCI DSS) [39] is one of the major requirements. The PCI DSS compliance can be achieved by striving for 6 card data protection objectives by practising 12 main requirements [37]. The 12-prescribed requirements are shown in Figure 3.

U.H. Rao et al. [73] highlight the importance of PCI DSS compliance, which can be achieved by audits at both internal and external levels and by complying with the organization’s technical and operational standards. The authors further highlight some of the key issues of non-compliance with PCI DSS requirements by various organizations and discuss different solutions for such issues of non-compliance, which are summarized in Table 2.

Most issues were resolved apart from the standardization of applications, which was not 100 percent successful. This issue remains as most organizations resist change. However, continuous awareness, monitoring, compliance, and auditing of the standards can help improve better protection of customers [73].

In the context of cyber-physical, critical and management information systems [46,47,49], to ensure their security and maintenance, the US government ensures that such systems are governed under specific federal law such as Federal Information Security Management Act (FISMA) [40]. Furthermore, to ensure the security of the aforementioned systems, FISMA assigned the National Institute of Standards and Technology (NIST) to devise certain controls. For fulfilling these requirements, NIST provided, implemented, and published FIPS 200 with seventeen control areas to ensure the well-being of such systems, as shown in Table 3 [37].

The federal agencies are required to ensure continuous system monitoring, and each agency receives an annual grade on FISMA compliance [37]. In the US, the health care industries [48], are regulated through Health Insurance Portability and Accountability Act (HIPAA) 1996 [44], which is administered by Health and Human Services (HHS) Department. HIPAA uses two rules to address security concerns, i.e., privacy and security rules. Entities that access information in HIPAA-compliant systems must incorporate the following points:Ensure Confidentiality, Integrity and Availability (CIA) of e-PHI during creation and communication.Identify and protect against potential threats.Protect against all possible impermissible uses and disclosure.Ensure compliance.

HHS guides entities on how to comply with the security requirements [37]. A flow diagram shows the compliance with requirements based on recommended processes in Figure 4.

In addition to HIPAA, the role of omnibus rules in reducing healthcare data breaches is also evident from the study [74]. The omnibus rules include enforcing various rules such as breach notification, privacy, and PHI. With the implementation of HIPAA omnibus rules [75], the study demonstrated a significant reduction in data breaches, including the prevention of expected 180 breaching incidents that could have affected millions of Americans’ data [74].

Finally, the literature also provides certain policies and best practices that must be incorporated to mitigate the impact of breaching incidents. These polices and best practices include: (a) a high percentage of budget must be allocated to IT sector for achieving information security [28]; (b) breaching event must be reported publicly to minimize the impact of subsequent breaches [61]; (c) according to K. Renaud study, the employees of health services must adhere to information security policies in order to keep patient information secure [76]; (d) The providers should adopt a security culture within the system; (e) the providers should organize training programs, and seminars for employee awareness about security policies and trends; (f) the employees should avoid sharing personal and organizational information on social media or other public platforms; (g) new policies should be directly introduced to employees upon login of their computer systems and employees had to declare that they have read and understood the newly implemented policy before entering their computer systems; (h) appoint a responsible person for the continuous surveillance of data logs [76]; and (i) The researchers demonstrated how negligence and failure of understanding a breach by the providers further increases its impact. Therefore, the providers must understand the breach to minimize the overall cost of security risks, data protection, breach mitigation, and incidence response [31].

### 4.3. Legal Obligations

Organizations must fulfil various legal obligations to mitigate the overall impact of data breach incidents. One of the major legal requirements for financial institutions is to comply with the Graham-Leach-Bliley Act 1999 [41], also known as the safeguard rule. The act provides a mechanism to secure customer information regarding its confidentiality and integrity. According to the study [37], the act binds financial institutions to develop and apply an information security program under the Federal Trade Commission (FTC) to ensure the following elements:Depute an employee to coordinate financial institutions’ information security programs.Identify all internal and external potential security risks to the PII and other confidential data using risk assessment.Design and implement controls against the identified risks and regularly monitor the effectiveness of the implemented controls.Select and maintain service providers capable of protecting customers’ data through continuous compliance.Oversee testing and monitoring results and adjust information security program accordingly [37].

Similarly, other regulations also impose legal obligations protecting organizations from different breaching incidents. Regulators worldwide are incorporating security and breach clauses in their laws due to increasing cyber incidents, and these laws differ among various jurisdictions. According to a comparative study conducted by Rita Heimes [13], about European General Data Protection Regulations (GDPR) [77], with Canadian [78], and US standards [43]. According to the author in [13], the GDPR protects the European citizens’ privacy and considers it citizens’ fundamental right to protect their data by giving them control over their data. GDPR also emphasizes data flow across each EU state by data controllers and processors involved. Article 32 of GDPR requires the implementation of organizational and technical measures to manage risk, such as:Encryption and hiding the identity of personal data.Ensure confidentiality, integrity, and availability of system and processes.Accountability and auditing of the organizational and technical measures for the security of processes.

Besides that, GDPR Article 33 states that data breach incidents should be notified without any delay to the supervisory authority; however, Article 34 requires one also to intimate the affected individual immediately if high-risk data leakage occurs by making it his/her fundamental right to know; any violation in this regard may result in penalties. The US data security and breach notification regulations are relatively industry-dependent or States-based. In the US, breach notification laws require that if the company promises the consumers to safeguard data, they are liable to certain penalties under the Federal Trade Commission Act [35,43]. The Canadian national privacy law: Personal Information Protection and Electronic Documents Act (PIPEDA) [78] frames rules for gathering, utilising, and disclosing personal data for commercial purposes. Subsequently, the Digital Privacy Act has amended a few clauses in PIPEDA, including data breach notification. Data breach notification is applied when the data are lost or accessed by an unauthorized entity. These existing laws generally do not specify security measures but help prevent breaches. Therefore, security professionals must work closely with privacy professionals, policymakers, legal counsel, and other regulators to build a secure and effective cybersecurity program.

Moreover, the recent Securities and Exchange Commission (SEC) regulation in the US forces firms to disclose breach events [34]. Furthermore, the author in [35] addresses and offers a broader framework to focus on different trade-offs and perceptions about regulations. According to the author, the policymakers for different industries in the US use two regulations, i.e., post and ante regulation. These regulations make industries consider investment in security and implementing standards and procedures. First, a post-regulation executes and states that if a data breach occurs and results in a customer’s data leakage, the said industry will be liable to certain penalties. Second, regulation is ante regulation, which states that each firm has to apply and maintain proper control for auditing procedures, such as PCI Standard [39], Fair Credit Reporting Act (FCRA) [79], and GLBA [41], etc. Another recommendation is the disclosure of data breaches to government agencies to take remedies on policy and procedure. However, the policy does not quantify every breach loss or avoid all breaches, but it improves the security level and culture of the firm. Thus, better security is possible if a fair amount of budget is allocated to undermining such breaches [35].

In addition, the researchers have also proposed certain recommendations to be enforced by laws to minimise the impact of data breaches. These recommendations include: (a) To encourage both public and private firms to enact the security breach state notification legislation to implement better security measures to protect consumer information [80]; (b) To force firms to disclose any breach publicly regardless of its severity and consequences. However, reputational cost does come with the public announcement of security breach [80]; (c) the legal framework should be reformed so that confidential information should properly be protected by both policy and legal obligations [36]; (d) to impose heavy penalties on violation of notification laws [19]; and (e) J.C. Ford et al. [19] propose a breach response life cycle which detects, notifies, and ultimately resumes the business as usual. The data breach response steps are shown in Figure 5.

Finally, the effectiveness to adhere with legal obligations is evident from these facts: (a) the United States Federal Trade Commission states that the introduction of data disclosure has reduced identity theft by 6 percent; therefore, other countries, such as the Australian government, also in the process of introducing data breach notification laws [63]; (b) According to a study [80], before the implementations of notification laws during the period from 2001–2006, a total of 92 breaching incidents had an impact of 0.07 percent on the firms’ market value. On the other hand, after the enactment of notification laws, despite an increased number of reported breaches during the period 2006–2008, a lesser impact of 0.05 percent on the firm’s market value was observed [80].

## 5. Analysis and Future Directions

In this section, we summarize and analyze the existing literature presented in the preceding sections that surveyed various aspects of data breaches. The “✓” sign is used in the following tables to indicate that a particular aspect is focused on in the referenced study.

Table 4 summarizes the studies on the impact of data breaches in various sectors. Similarly, Table 5 shows the body of research identifying various reasons for the loss of information.

The reasons for such impact include identity theft, fraud, unintended exposure by both customers and employees, negligence on the part of employees, loss of trust/reputation, less allocated IT staff, low-security measures, and low budget allocated towards the IT sector. These Table 4 and Table 5 highlight the reasons that contribute to the loss of customers and information in various sectors. We observe that many studies target the issue of data breaches; however, the literature lacks a comprehensive study which considers all reasons that increase the overall impact of breaching incidents. This study also indicates that very few researchers discuss data breaches and their impact on the critical sector. Consolidated approaches that trace and counter data breaches is a subject yet open, preferably on policies, laws, and regulations on national as well as international level. Individuals, as well as states, use these breaches as a tool for warfare and cyber-attacks. For example, the US formulated the cyber-attack policy and called it an act of terrorism and an attack on national security. This policy is also not free from criticism [33]. To minimize such attacks, proper policies and laws formation is the key to future work, which identifies new methods, techniques, and frameworks that continuously assess system breaches, especially in Critical Information Sectors and Health Management Systems.

Table 6 shows the existing preventive measures in literature, which must be adopted to mitigate the impact of data breaches. The literature provides various preventive measures, such as proposed models and techniques, standards, policies, procedures, best practices, and legal obligations. The existing body of research proposes many models and techniques for data breach prevention but the literature lacks a comprehensive framework. Using common security mechanisms for data security in parts is also not sufficient specially for use case, such as smart cities, industry 4.0, massive IoT data, etc. Following international standards such as ISO 27001, PCI DSS, etc., and other mandatory legal obligation such as GDPR greatly help in the protection of data and reducing data breach incidents; however, obtaining compliance can be challenging and difficult for every organization.

Finally, Table 7 summarizes the literature study presented in this paper. It organizes the studies on the impact of a data breach on various sectors, financial assets, and people; furthermore, the table provides the effect of such an impact. Additionally, studies on various reasons contributing to data breaches, along with mitigation strategies to minimize such incidents, are highlighted. The table also shows studies on follow-up strategies for continuous compliance and improvement to avoid such breaching incidents.

These results are elaborated by presenting data in graphs shown in the following figures. Figure 6 plots the number of studies that identify various possible reasons for data breaches in various sectors. Similarly, Figure 7 presents a graphical representation of all preventive measures highlighted in the literature. This analysis shows that identity theft is the most common reason for a data breach, which demands more robust identity management solutions.

The available literature provides a broad picture of preventive measures used to prevent data breaches (Table 7 and Figure 7); however, we do not find any comprehensive solution to integrate all these preventive measures. We thereby highlight the need for a comprehensive framework incorporating preventive measures per the sector’s needs. Similarly, there is no comprehensive and complete set of framed principles, such as standards, policies, procedures, and best practices which are completely followed among various sectors. The reason may be negligence on the part of both employees and customers. Moreover, there is a lack of awareness and training for security programs, ultimately leading to non-compliance with aforesaid principles.

Furthermore, the issue of non-compliance with legal obligations is also the main reason behind major breaching events. These legal obligations are not fulfilled due to reasons such as a lack of cohesive legal framework and non-compliance with notification laws, federal laws, and state laws; however, to deal with this issue, the research community, academia, and policymakers must frame such a comprehensive law that is applicable both at a local and global scale. The work can be considered similar to establishing an international body responsible for regulating and ensuring that all sectors must abide by the framed legal principles. Every sector should be a member of this regulatory body. The regulatory body should also constitute assessment bodies within each state or country. Finally, a follow-up strategy must also be incorporated to ensure continuous monitoring, auditing, and assessment of compliance with framed principles and legal obligations.

### Future Directions

We identify three main future directions based on these observations and literature study.There is a need of a generic framework for the continuous risk assessment of data breaches and their overall impact so that appropriate risk treatment (mitigation, avoidance, insurance or acceptance) can be done. A preliminary conceptual framework could include four major phases (shown in Figure 8). In the first phase, potentially critical data which could be breached and can impact the organization is ***identified*** using data governance approaches. In phase 2, the organization ***assesses breach risk*** in the processes, functions, and systems. Next, in phase 3, based on the prioritized breach risks, the organization ***implements controls***, policies, procedures, standards, and regulations. Last, in phase 4, the organization ***monitors the effectiveness*** of the implemented controls on its processes, functions, and systems. Finally, the continuity of the model is ensured by going through all phases in case of any lapse or any addition to data, assets, actors, and processes.Different methods to quantify and calculate breach cost are presented in the literature, which somehow calculates the cost associated with data breaches. However, these cost calculators and quantifiers are based on breach windows identified by organizations. It is a matter of great concern that the organization, large or small, hides the issues caused by breaches, which are not disclosed publicly. Therefore, these cost estimation calculators do not calculate the real cost of a breach. Thus, there is a need for such calculators and quantifiers that calculate the exact cost associated with breaching incidents by analyzing both notified and associated hidden breaches resulting from notified breaches.A possible solution to reduce the data breach’s impact is to enforce an effective consent framework. In this framework, data access, deletion, and usage are only granted to legitimate owners. Ever-changing trends, environments, and technology bring new challenges and privacy issues. Therefore, a privacy framework may also be designed to test system applications continually and log each installed application activity, including permission grants and access violations, formulate policies accordingly, and block such applications that escalate and creep any privilege. Moreover, any unauthorized activity should be notified to legitimate users, and any unauthorized data movement, if so detected, may be self-destroyed or encrypted.

## 6. Conclusions

The approach of this study is more comprehensive than the existing literature on the impact of data breaches, its possible reasons, mitigation and follow-up strategies. The study highlights certain limitations, such as a lack of technical and legal frameworks, security awareness and training, and regulatory and assessment bodies. Thus, there is an immediate need on the part of researchers, academia, and industry to address these limitations. Therefore, according to our knowledge, this study provides a platform for researchers to address data breach incidents. Moreover, the study also highlights a need to provide a continuity model to mitigate the risk of data breaches that must be followed by various sectors of smart cities.

In addition, the study also highlights that the security and privacy of data is not critical only in the smart cities context, but small organizations can also be affected financially by data breach incidents, whereas reputation is a major concern for the large enterprises. We have also observed that the compliance to standards and regulations can play a major role in the prevention of data breach incidents. While ensuring compliance to complex standards and regulations can be challenging, and some times counter productive, for SMEs and public organizations, due to their less skilled staff and budgetary constraints, cloud based managed services could allow such organizations to obtain compliance by day-zero.

Finally, this study also reflects the importance of people as an asset to the organization. For instance, a high-profile personal or chief security information officer or any other employee associated with high access levels must be protected against physical lapses. Furthermore, to overcome the incidents, it is suggested that strategies such as layered security or defence in depth, change management, separation of duties, etc., must also be incorporated by sectors to enhance the security of information, systems, and other high-value assets. 

## Figures and Tables

**Figure 1 sensors-22-09338-f001:**
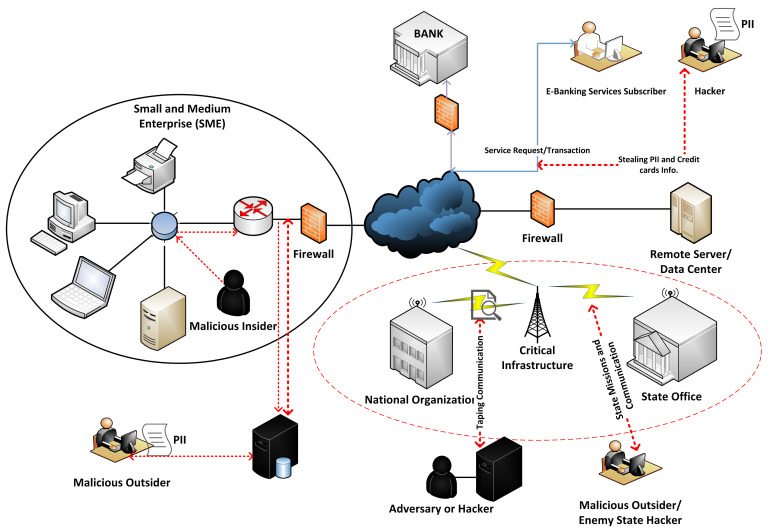
Large attack surface exposed by the connected infrastructure cause data breaches.

**Figure 2 sensors-22-09338-f002:**
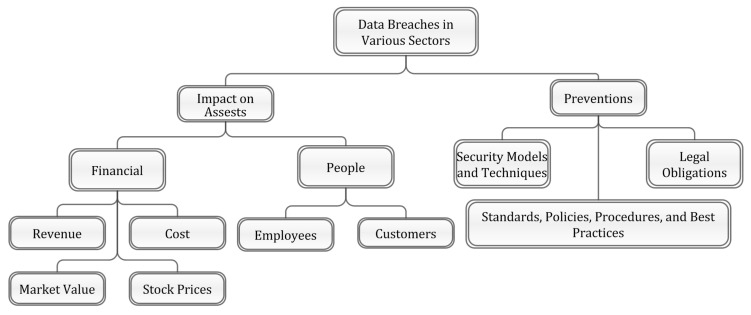
Data Breaches’ Taxonomy.

**Figure 3 sensors-22-09338-f003:**
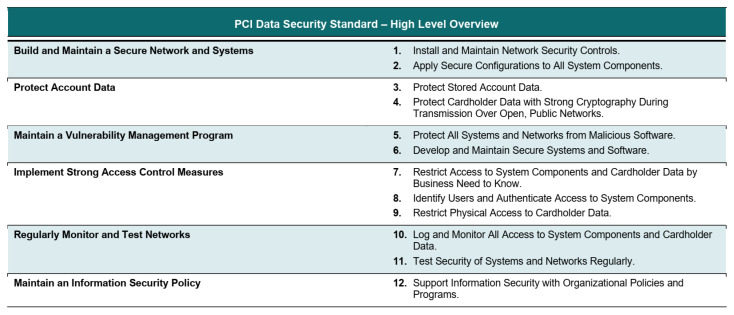
PCI DSS security goals and 12 principal requirements for the protection of cardholder data [39].

**Figure 4 sensors-22-09338-f004:**
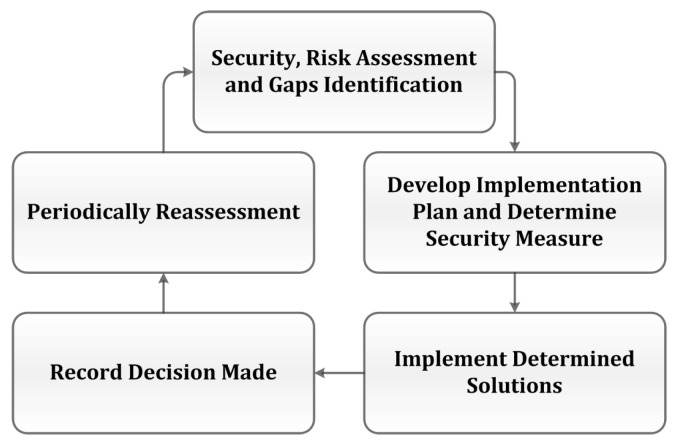
HSS Recommended Compliance Process.

**Figure 5 sensors-22-09338-f005:**
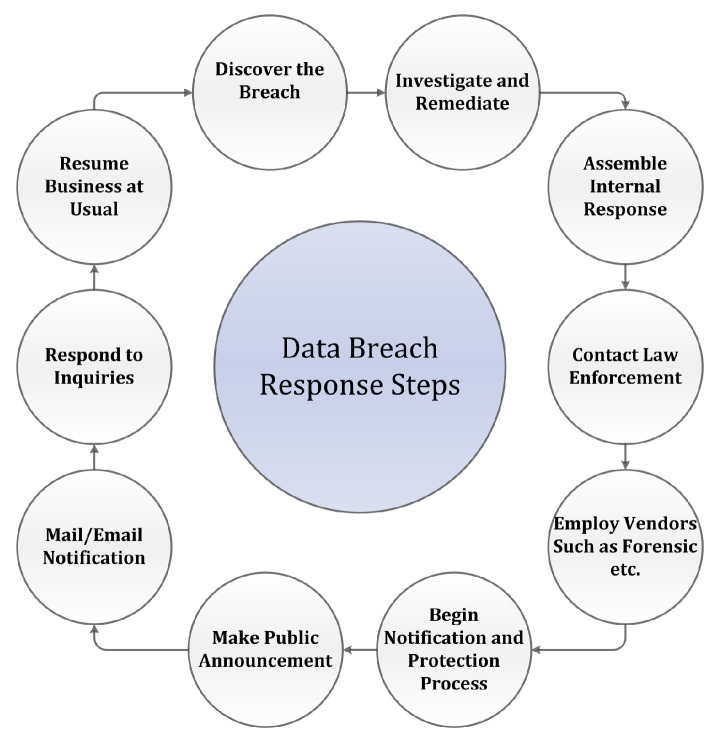
Data breach response steps.

**Figure 6 sensors-22-09338-f006:**
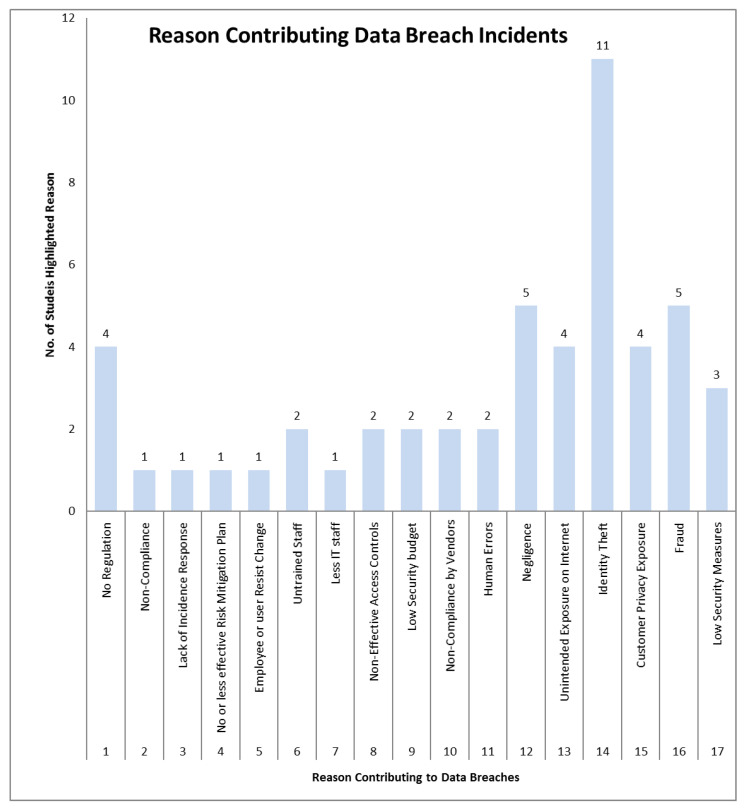
Graphical view of reasons contributing to data breaches.

**Figure 7 sensors-22-09338-f007:**
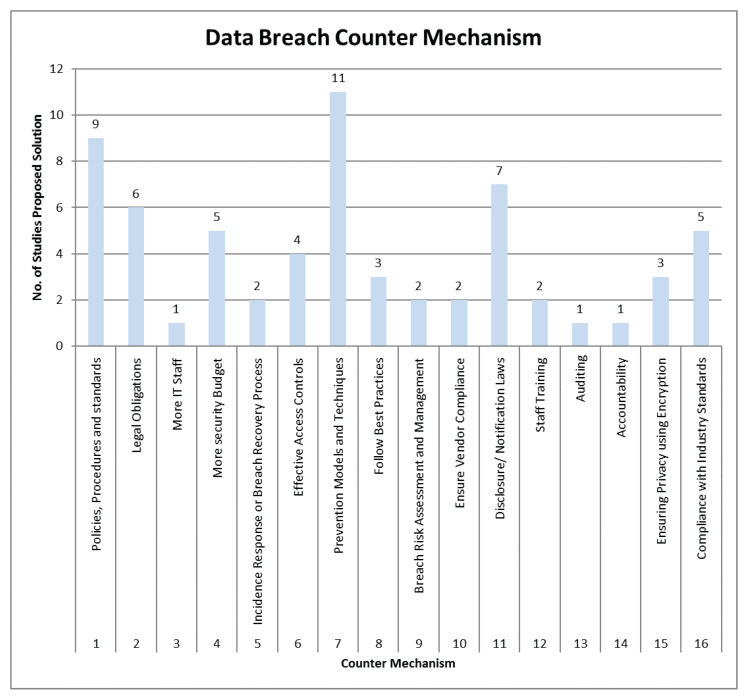
Graphical view of all counter measures to data breaches.

**Figure 8 sensors-22-09338-f008:**
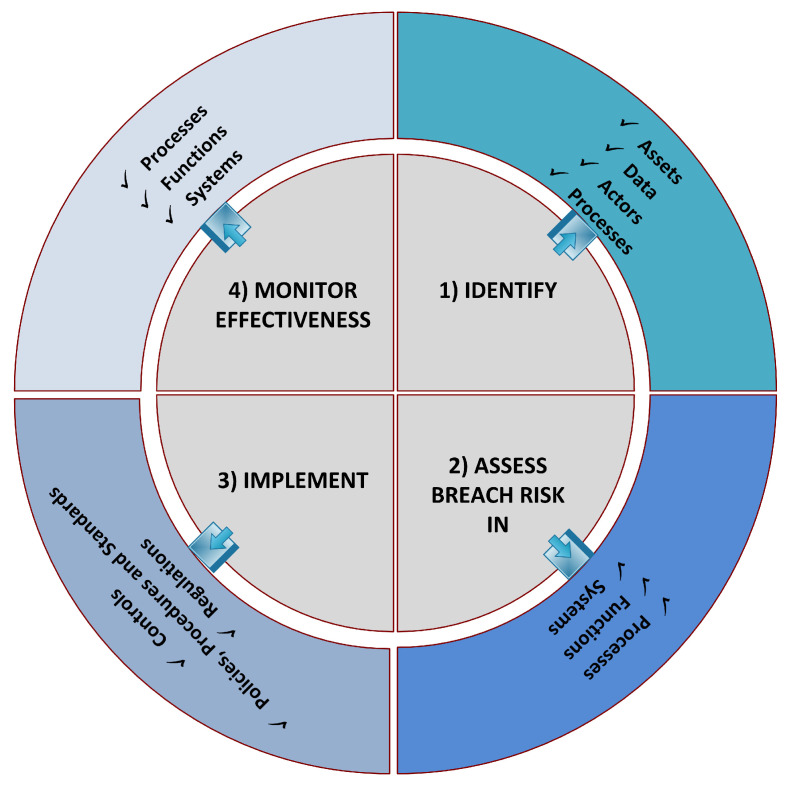
Continuous data breach risk assessment framework.

**Table 1 sensors-22-09338-t001:** Domains of ISO 27001 and distribution of 93 security controls in each domain.

1.	Organizational Controls (37)	2.	People Controls (8)
3.	Physical Controls (14)	4.	Technological Controls (34)

**Table 2 sensors-22-09338-t002:** Issues related to non-Compliance of PCI DSS requirements and their solutions [73].

Requirement	Issues	Solutions
8	Monthly Premium Payment	Unique Agent ID for Every Organization.
3,4	Plaintext Email Attachment	Use of Application to generate encryption and decryption code for every card.
7	Accessibility of Card Holder User Data	Implementation of Access Control Policy and Regular Review of Control Policy.
2	Leaking of Encryption Keys by Former Employees	Fixing Administrative Controls and Continual Key Update.
6,11	Lack of Standardization of Used Applications	Using Validated and Certified Third-Party Applications.
10	Lack of Mechanism to Monitor and Analyze Logs	Acquiring tools for Continual Monitoring and Analyzing of logs.
12	Lack of IS policy propagation and Awareness	Using NIST 2003 mandatory video sessions to carry out businesses without losing reputation.
Other	Lack of Effective Vulnerability and Pen Testing	Experienced Involvement and adapting to new PCI-DSS 3.0 requirements.
	Lack of Identification and Management of CIA issues	Continuous Auditing to ensure information.
	Continued Sustenance of PCI-DSS Compliance	Continuous effort to stay compliant.

**Table 3 sensors-22-09338-t003:** 17 Control Areas of NIST FIPS 200.

1.	Identification and Authentication	2.	Access Control
3.	Awareness and Training	4.	Auditing and Accountability
5.	Assessment and Certification	6.	Incident Response
7.	Contingency Planning	8.	Configuration Management
9.	Physical and Environment Protection	10.	System and Information Integrity
11.	Maintenance	12.	Planning
13.	Risk Assessment	14.	System and Communication Protection
15.	System and Service Acquisition	16.	Media Protection
17.	Personnel Security		

**Table 4 sensors-22-09338-t004:** Contribution of various studies on the impact of data breaches. The impact on the People category can result in the loss of Trust, Reputation, Turnover and Confidence, which are represented by **Tr**, **R**, **Tu**, and **C**, respectively.

	SECTOR					FINANCIAL				PEOPLE
	Health-Care	Critical Cyber-Physical	Gaming	IT/ SMEs	Financial	Cost	Rev-Enue	Market Value	Stock Price	Employee-Customer
[18]				✓				✓		R, C
[19]						✓				
[28]				✓		✓		✓	✓	
[30]	✓	✓		✓	✓					C
[31]		✓				✓				R, C
[32]			✓			✓		✓		R, C
[33]				✓		✓	✓		✓	
[34]				✓		✓		✓		
[36]		✓				✓	✓	✓	✓	
[52]	✓					✓				C
[53]	✓									Tr, R, C
[55]			✓			✓			✓	
[58]				✓		✓	✓			
[59]						✓				
[60]						✓				C
[61]						✓		✓	✓	
[80]								✓		

**Table 5 sensors-22-09338-t005:** Reasons contributing to the loss of information.

	Identity	Fraud	Unintended	Negligence of	Lack of	Lack of	Budget
	Theft		Exposure	Employees	Skilled Staff	Security	Constraints
[18]							✓
[19]	✓						
[28]	✓	✓					✓
[30]			✓	✓			
[31]	✓	✓		✓			
[32]	✓	✓					
[33]	✓						
[52]	✓	✓	✓	✓		✓	
[53]	✓		✓	✓	✓	✓	
[61]	✓	✓			✓		
[63]	✓						
[70]			✓			✓	
[76]	✓		✓	✓			
[80]	✓						

**Table 6 sensors-22-09338-t006:** Table presents various preventive measures focused in the literature. Well known international standards or regulations are scribed, whereas ✓ is used for local or national obligation.

	Proposed Model and Techniques	Regulations, Policy and Procedures	Standards	Best Practices	Legal Obligations
[13]					GDPR
[19]					✓
[31]				✓	
[32]	✓				
[34]					✓
[35]			PCI DSS		GLBA * FCRA *
[36]		✓			✓
[37]			PCI DSS		FISMA * HIPAA * GLBA *
[53]	✓	✓		✓	
[58,59]	✓				
[61]		✓			
[63]					✓
[64,65,66,67]	✓				
[68]	✓				
[70]	✓				
[73]			PCI DSS		
[74]					HIPAA *
[76]		✓		✓	
[80]					✓

* local/state obligation.

**Table 7 sensors-22-09338-t007:** Data breaches’ impact and prevention strategies.

	Impact Category	Effects	Reasons	Treatment	Mitigation	Follow-Up
**Data Breaches**	**Sectors** [46,47,48,49,50]	Critical/Cyber-physical System [35,36,56,57]	No Regulation [13,35,36,80] Non-Compliance [19] Lack of Incidence Response [31] No or less effective Risk Mitigation Plan [81] Resist Change [73] Untrained Staff [53,76] Less IT staff [53] Non-Effective Access Controls [68,73] Low Security budget [18,28] Non-Compliance by Vendors and Service Providers [31,53]	Incidence Response or Breach Recovery Process [19,30] Disclosure/Notification Laws [13,19,32,33,51,60,77]	Policies, Procedures and standards [33,35,36,37,53,61,73,74,76] Legal Obligations [19,34,35,37,63,80] More IT Staff [53] More security Budget [18,27,28,34,35] Incidence Response or Breach Recovery Process [19,31] Accountability Effective Access Controls [37,68,73,81] Implement Proposed Models and Techniques [32,33,53,58,59,64,65,66,67,68,70] Follow Best Practices [31,53,76] Breach Risk Assessment and Management [38,81] Ensure Vendor and Provider Compliance [31,53]	Continuous Auditing [13,33,53,73] Continuous Monitoring [37,38,40,73,76] Continuous Compliance [37] Breach Risk Reassessment [38,81]
		Healthcare [31,52,53,70,74]				
		Gaming [32,55]				
		IT Firms/ other Businesses (Large/Small) [18,30]				
	**Financial** [56]	Market Value [34,61]	No Regulation [13,35,36,80] Non-Compliance [19] Lack of Incidence Response [31] No or less effective Risk Mitigation Plan [81]	Incidence Response or Breach Recovery Process [19,30] Disclosure/Notification Laws [13,19,32,33,51,60,77]	Disclosure/Notification Laws [13,19,33,34,53,63,80] Implementation of Economic and Financial Models [33,58,59] Cost Calculators and Estimators [33,58]	
		Cost [19,58,59,60]				
		Revenue [58]				
		Stock Prices [61]				
	**People** [30,31,52]	Company data/ information [18,28,52,53]	Human Errors [30,31] Negligence [30,31,52,53,76] Unintended Exposure [30,52,53,76]	Incidence Response or Breach Recovery Process [19,30] Disclosure/Notification Laws [13,19,32,33,51,60,77]	Staff Training [53,76] Auditing [53] Accountability [53]	
		User name and Passwords [32,52,55]				
		Loss of Customer [19,28,30,31,32,33,52,61,63,76,80]	Identity Theft [19,28,31,32,33,52,53,61,63,76,80] Customer Privacy Exposure [30,52,53,76] Fraud [26,29,30,51,60] Less IT Staff [53] Low Security Measures [52,53]		Disclosure/ Notification Laws [13,19,33,34,53,63,80] Incidence Response Plan [19,31] Ensuring Privacy of data using Encryption [13,53,73] Compliance with Industry Standards [37,38,39,40,44]	
		Loss of Reputation and Trust [20,28,32,52,61,80]				

## Data Availability

Not applicable.

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
