# Peer review of "Getting Smarter about Smart Cities: Improving Data Security and Privacy through Compliance"

_sensors, 2022, doi:10.3390/s22239338_

Round 1
Reviewer 1 Report
The authors of this article highlight the importance of data breaches and issues related to information leakage incidents. In particular, the impact of data breaching incidents and the reasons contributing to such incidents affect the citizens’ well-being.
Paper's Title
The paper's title gives the impression that the discussion will focus on smart cities' adopted technologies and how data breaches are committed in such cities. But the paper discusses the data breaches that happen in any city, not specifically smart cities; hence I found the article's title misleading. Moreover, the title indicates "Improving data security and privacy through legal compliance," but the paper does not mention how to achieve this. The paper, in its majority, presents an Adhoc (not systematic) literature review of data breaches, preventive actions, procedures & policies, and standards.
Comments:
1. Figure 1
This figure is not explained; it's better for the reader to read an explanation of this figure
2. Page 2, line 47:
The survey cited in [18] is an old study. Can you cite some more recent studies in this regard
3. Page 4, line 110:
The taxonomy provided is superficial. You did not discuss how the taxonomy has been developed; referring to 2-3 references to build a taxonomy is insufficient. Read the following reference to see how the data breaches taxonomy is developed and how your taxonomy is compared to this one should be explained.
http://eprints.covenantuniversity.edu.ng/15593/
4. Page 6, lines 217-219:
Are you sure of these numbers? $418,000 is a small amount. Anyways, you are using Ponemon’s studies of 2015 and 2019; I suggest updating your data as the report of 2022 is already published
5. Page 9, line 374:
A newer version of ISO27001 has been released in 2022. If you cite it, your article will be more contemporary. The newer version has 93 controls rather than 114 controls in version 2013.
6. Page 9, line 381:
2013's version has 114 control, not 39! The newer version of 2022 has 93 controls. Revise your text and numbers in this regard
7. Figures and Tables
The figures and tables' sources should be cited in the caption. Most of the figures are not new, and the literature has similar or better illustrations, so I suggest using such figures and citing the reference—for instance, Figure 4 PCI-DSS Compliance process where the literature and the related websites have more attractive illustrations
8. Page 15, Table 4:
You mean the impact of data breaches on organizations' assets.
According to Robert Layton and Paul A.Watters [31], an organization comprises financial assets and people. Here financial assets are further divided according to the literature on four parameters, namely: (a) cost, (b) revenue, (c) market value, and (d) stock price. Next, the asset people are further classified as (a) customers and (b) employees, which impact the organization in terms of loss of trust, reputation, turnover, and confidence.
Why did you add the sector to this table? I may accept adding the sector to give a wider view. But still, I expect to see a column that shows what dimension has been impacted by the organization discussed in each reference (e.g., loss of trust, reputation, turnover, and confidence.)
9. Page 16, Table 5:
In table 4, you mentioned four references that have discussed the impact of data breaches on people. While in table 5, we have 14 references that discuss the people loss (employees). Why not add the remaining ten references to table 4?
10. Page 17, Table 6:
Here, you labeled each reference and the type of preventive measure it adopted. This provides less helpful information to the reader. I suggest adding the adopted measure per reference; for example, when you say that reference 33 adopted standards, which standard is adopted?
11. Page 21, line 592:
Presenting the new preliminary conceptual framework for the Continuous Data Breach and Risk Assessment in the future work section is useless. You did not adequately present the framework. So, you either discuss it in detail in the paper or mention the need for such a framework in the future work section. What you did is a hewn and unclear presentation of the framework.
Author Response
REVIEWER 1
First of all, we would like to thank the reviewer for useful suggestions to improve the quality of our manuscript. In the revised version, we have addressed these comments and made the appropriate changes in the manuscript, as reported below. For the reviewers’ convenience, we have highlighted the changes in red.
Reviewers Comment 1
Paper's Title: The paper's title gives the impression that the discussion will focus on smart cities' adopted technologies and how data breaches are committed in such cities. But the paper discusses the data breaches that happen in any city, not specifically smart cities; hence I found the article's title misleading. Moreover, the title indicates "Improving data security and privacy through legal compliance," but the paper does not mention how to achieve this. The paper, in its majority, presents an Adhoc (not systematic) literature review of data breaches, preventive actions, procedures & policies, and standards.
Response
The title is slightly changed by removing the word “legal”, as we find in the preventive techniques that data breaches can be minimized by ensuring compliance with various standards and regulations. The literature in Table 6 is aggregated to emphasize that compliance with standards and regulations result in better security and fewer cases of data breaches. Surveys mentioned in the paper also indicate a downward trend in data breaches when standards and regulations are followed.
Reviewers Comment 2
Figure 1 This figure is not explained; it's better for the reader to read an explanation of this figure.
Response
Explanation added.
Reviewers Comment 3
Page 2, line 47: The survey cited in [18] is an old study. Can you cite some more recent studies in this regard
Response
The latest surveys by IBM and Statistica (Identity Theft Resource Center) have been added.
Reviewers Comment 4
Page 4, line 110: The taxonomy provided is superficial. You did not discuss how the taxonomy has been developed; referring to 2-3 references to build a taxonomy is insufficient. Read the following reference to see how the data breaches taxonomy is developed and how your taxonomy is compared to this one should be explained.
http://eprints.covenantuniversity.edu.ng/15593/
Response
Figure 2 on Page 10 gives an impression that the presented taxonomy is weak; however, it is discussed and enriched in the subsequent sections and later summarized in tables 4,5,6 and 7 with a sufficient number of references.
After reading the abstract of the shared thesis link, it looks quite interesting. However, it is not accessible to download. It is either not published or available publicly.
Reviewers Comment 5
Page 6, lines 217-219: Are you sure of these numbers? $418,000 is a small amount. Anyways, you are using Ponemon’s studies of 2015 and 2019; I suggest updating your data as the report of 2022 is already published
Response
Thanks for the correction. It is indeed a typo. The average cost was 4.18 million, which becomes $4,180,000. As you suggested, the latest figures from the 2022 report of IBM have been added.
Reviewers Comment 6
Page 9, line 374:
A newer version of ISO27001 has been released in 2022. If you cite it, your article will be more contemporary. The newer version has 93 controls rather than 114 controls in version 2013.
Response
This is indeed a good suggestion to update the paper with the newly released version. Changes in the text, Table 1 and references are made.
Reviewers Comment 7
Page 9, line 381: 2013's version has 114 control, not 39! The newer version of 2022 has 93 controls. Revise your text and numbers in this regard
Response
Revised according to ISO 27001:2022
Reviewers Comment 8
Figures and Tables: The figures and tables' sources should be cited in the caption. Most of the figures are not new, and the literature has similar or better illustrations, so I suggest using such figures and citing the reference—for instance, Figure 4 PCI-DSS Compliance process where the literature and the related websites have more attractive illustrations
Response
We have updated the text with reference to PCI 4.0 (latest release), replaced figure 3 with the figure-table from source PCI 4.0 and cited the reference in the caption. Moreover, Figure 4 has been removed because it would be redundant otherwise.
Reviewers Comment 9
Page 15, Table 4: You mean the impact of data breaches on organizations' assets.
According to Robert Layton and Paul A.Watters [31], an organization comprises financial assets and people. Here financial assets are further divided according to the literature on four parameters, namely: (a) cost, (b) revenue, (c) market value, and (d) stock price. Next, the asset people are further classified as (a) customers and (b) employees, which impact the organization in terms of loss of trust, reputation, turnover, and confidence.
Why did you add the sector to this table? I may accept adding the sector to give a wider view. But still, I expect to see a column that shows what dimension has been impacted by the organization discussed in each reference (e.g., loss of trust, reputation, turnover, and confidence.)
Response
Keeping sector gives a better understanding of the context; however, we have found that there are few references which do not discuss the impact of breach even though they focus on the data breach topic. We have removed those references from Table 4. Moreover, as suggested, we have added alphabets representing Loss of Trust, Reputation, Turnover and Confidence (Tr, R, Tu, C).
Reviewers Comment 10
Page 16, Table 5: In table 4, you mentioned four references that have discussed the impact of data breaches on people. While in table 5, we have 14 references that discuss the people loss (employees). Why not add the remaining ten references to table 4?
Response
Loss of customers is an impact already indicated in Table 4; therefore, it is removed from this table. This also addresses the impression that these 14 references discuss people's loss. In fact, table 5 has been updated to indicate the loss of information due to various reasons.
Reviewers Comment 11
Page 17, Table 6: Here, you labeled each reference and the type of preventive measure it adopted. This provides less helpful information to the reader. I suggest adding the adopted measure per reference; for example, when you say that reference 33 adopted standards, which standard is adopted?
Response
The suggestion is useful, but while trying to do so, the table gets filled with lots of text, especially in the regulations column. US regulations (HIPAA, FISMA, etc.) are famous and much discussed in the literature, which can be shown with abbreviations; however, regulations from Malaysia, Australia and other countries cannot be abbreviated. Therefore, we have adopted a midway approach to such regulations. International standards are well known and are written in the table. We hope that this will improve readability.
Reviewers Comment 12
Page 21, line 592: Presenting the new preliminary conceptual framework for the Continuous Data Breach and Risk Assessment in the future work section is useless. You did not adequately present the framework. So, you either discuss it in detail in the paper or mention the need for such a framework in the future work section. What you did is a hewn and unclear presentation of the framework.
Response
We cannot explain the conceptual framework as it is only a future work idea at this stage. This can also be removed, but we find gaps in the existing literature therefore, we feel that we should mention its need as suggested by the reviewer. The text in the manuscript is revised accordingly.
Reviewer 2 Report
Please see the attached report.

Author Response
REVIEWER 2
First of all, we would like to thank the reviewer for useful suggestions to improve the quality of our manuscript. In the revised version, we have addressed these comments and made the appropriate changes in the manuscript, as reported below. For the reviewers’ convenience, we have highlighted the changes in red.
Reviewers Comment 1
In my view, the literature review can further be improved by referring to the latest literature such as -- - -
- "Motorcycle Ban and Traffic Safety: Evidence from a Quasi-Experiment at Zhejiang, China"--
- "Smart City Construction and Management by Digital Twins and BIM Big Data in COVID-19 Scenario"--
- "Dynamical community detection and spatiotemporal analysis in multilayer spatial interaction networks using trajectory data"-
- -"C2FDA: Coarse-to-Fine Domain Adaptation for Traffic Object Detection"-
- -"Diversified Personalized Recommendation Optimization Based on Mobile Data"--
- "The continuous pollution routing problem"--
- "Electric vehicle routing problem: A systematic review and a new comprehensive model with nonlinear energy recharging and consumption"--
- "Data Collection in MI-Assisted Wireless Powered Underground Sensor Networks: Directions, Recent Advances, and Challenges"--
- "A Structural EvolutionBased Anomaly Detection Method for Generalized Evolving Social Networks"--"PPO-CPQ: A PrivacyPreserving Optimization of Clinical Pathway Query for E-Healthcare Systems"--
- "Improving high-impact bug report prediction with combination of interactive machine learning and active learning"--
- "Data Quality Matters: A Case Study on Data Label Correctness for Security Bug Report Prediction"--
- "A new IMUaided multiple GNSS fault detection and exclusion algorithm for integrated navigation in urban environments"--
- "An Indirect Eavesdropping Attack of Keystrokes on Touch Screen through Acoustic Sensing"--
- "Continuous Authentication Through Finger Gesture Interaction for Smart Homes Using WiFi"- -
- "Forecasting Urban Land Use Change Based on Cellular Automata and the PLUS Model"--
- "A Few Shot Classification Methods Based on Multiscale Relational Networks"--
- "Data-driven dynamic harmonic model for modern household appliances"--
- "Efficient Medical Big Data Management With Keyword-Searchable Encryption in Healthchain"--
- "A piecewise probabilistic harmonic power flow approach in unbalanced residential distribution systems"--
- "Modeling Relation Paths for Knowledge Graph Completion".
Response
We have added the latest relevant literature, such as ISO 27001:2022, 27002:2022, PCI DSS 4.0, Cost of a Data Breach (2022), and ITRC Data Breach Analysis H1 2022, to address the reviewer’s concern.
Reviewers Comment 2
Authors should thoroughly check their manuscript for typos and grammatical mistakes.
Response
We have proofread the manuscript again to remove existing typos and mistakes.
Reviewers Comment 3
Conclusion can be improved by highlighting the implications and expansion of this study to different landscapes.
Response
The conclusion has been revised
Reviewers Comment 4
I would like to see major improvements in the final form of this paper in terms of more discussion on preventive techniques.
Response
There is already a full section 4 on preventive techniques, which is discussed in detail. We have also added more discussion in the analysis of Table 6.
Round 2
Reviewer 2 Report
The authors have addressed all my comments except the comment related to the literature review is not taken into account. In any case, I can see that this manuscript is sufficiently improved and may be accepted.